# Critical Role of Etching Parameters in the Evolution of Nano Micro SLA Surface on the Ti6Al4V Alloy Dental Implants

**DOI:** 10.3390/ma14216344

**Published:** 2021-10-23

**Authors:** Pankaj Chauhan, Veena Koul, Naresh Bhatnagar

**Affiliations:** 1Mechanical Engineering Department, Indian Institute of Technology, Delhi 110016, India; dr.prachichauhan@gmail.com; 2Centre for Biomedical Engineering, Indian Institute of Technology, Delhi 110016, India; veenak_iitd@yahoo.com

**Keywords:** dental implants, osseointegration, titanium alloy, acid etching, microtopography, nanotopography

## Abstract

The surface of dental implants plays a vital role in early and more predictable osseointegration. SLA (sandblasted large grit and acid-etched) represents the most widely accepted, long-term clinically proven surface. Primarily, dental implants are manufactured by either commercially pure titanium (CP-Ti) or Ti6Al4V ELI alloy. The acid etch behavior of CP-Ti is well known and its effects on the surface microstructure and physicochemical properties have been studied by various researchers in the past. However, there is a lack of studies showing the effect of acid etching parameters on the Ti6Al4V alloy surface. The requirement of the narrow diameter implants necessitates implant manufacturing from alloys due to their high mechanical properties. Hence, it is necessary to have an insight on the behavior of acid etching of the alloy surface as it might be different due to changed compositions and microstructure, which can further influence the osseointegration process. The present research was carried out to study the effect of acid etching parameters on Ti6Al4V ELI alloy surface properties and the optimization of process parameters to produce micro- and nanotopography on the dental implant surface. This study shows that the Ti6Al4V ELI alloy depicts an entirely different surface topography compared to CP-Ti. Moreover, the surface topography of the Ti6Al4V ELI alloy was also different when etching was done at room temperature compared to high temperature, which in turn affected the behavior of the cell on these surfaces. Both microns and nano-level topography were achieved through the optimized parameters of acid etching on Ti6Al4V ELI alloy dental implant surface along with improved roughness, hydrophilicity, and enhanced cytocompatibility.

## 1. Introduction

Dental implants were introduced by Brånemark in 1960 for the replacement of missing teeth [1]. The well documented long term clinical result favors titanium and its alloy as the gold-standard material for dental implant application [2,3,4]. Titanium exhibits the best combinations of properties like strength, corrosion resistance, and biocompatibility as desirable for the bone–implant application. According to the ASTM standard, six types of titanium are available for biomedical implant applications including four grades of commercially pure titanium and two alloy forms (Ti6Al4V and Ti6Al4VELI grade). CP-Ti is an unalloyed pure form of titanium that contains only traces of other elements (i.e., carbon (C), nitrogen (N), oxygen (O), and iron (Fe). From Grades 1–4, there is an increase in oxygen content, which improves the mechanical properties of titanium. However, the alloy exhibits better mechanical properties than all grades of CP-Ti [3]. Ti6Al4V and Ti6Al4V-ELI alloys have a biphasic composition consisting of alpha and beta phases. Aluminum in these alloys act as an alpha phase stabilizer and vanadium acts as a beta phase stabilizer. Ti6Al4V ELI has a low concentration of interstitial elements O and C, which improves ductility compared to Ti6Al4V [3].

In the dental implant market, most commercial dental implants are made up of either Grade 4 CP-Ti or Ti6Al4VELI alloys. CP-Ti possesses good corrosion resistance while alloy has more strength (ultimate tensile, yield, and fatigue strength) required for long-term performance [3,4]. Therefore, depending on the clinical situations like lesser space or reduced bone quantity, which need narrow-diameter implants or where the occlusal forces are high, a dental implant must be manufactured from high strength material to prevent fatigue failure. Although both CP-Ti and Ti alloys are biocompatible, their surfaces are bioinert and it almost takes 3–6 months for them to osseointegrate with the surrounding bone. However, with the increase in the horizon of implant dentistry, there is a need for implant placement in less ideal bone conditions. Moreover, the introduction of newer protocols requires either immediate implant placement after extraction or immediate or early loading after implant placement. Therefore, there is a need to make the implant surface bioactive or conducive to accelerate the process of osseointegration. In the literature, various methods of surface modifications have been used to modify the titanium dental implant surfaces [5,6]

The sandblasting and acid etching process has been used commonly for dental implant surface modification to produce the SLA surface and clinically, these surfaces have been used actively for the past 3–4 decades. This combination introduces both macro- and microroughness necessary for the early osseointegration process [7]. Both sandblasting and acid etching are subtractive methods of surface modification. Acid etching creates pits or grooves on the metal surface by the process of selective corrosion [4]. The exact topography and dimensions of pits and resulting surface roughness depend on the types and combinations of acids used, their concentration, temperature, and duration of treatment. Besides dual roughness, the acid etching provides benefits like smoothening of sharp edges produced by sandblasting, removes any residual embedded blast media, contaminated oxide layer from the titanium surface, and provides modified chemistry, resulting in a bioactive surface. Various studies have documented the role of the etched surface over the machined and blasted surface [8].

Moreover, the reason for using a combination of methods is that the blasting procedure hypothetically achieves an optimal roughness for mechanical fixation, whereas the additional etching introduces micron level topography, which modulates the early host tissue response required for faster osseointegration. The resulting surface has an improved potential for protein adhesion, considered to be important for the early bone-healing process [9]. Etched surfaces have been reported for more bone apposition [10] and enhanced interfacial strength as measured by removal torque or push-out tests compared to machined surfaces [11]. Nanotechnology for dental implant surfaces has evolved in the recent past, and its success has been proven both in vitro and in vivo [12,13,14,15]. Both micro and macro thread geometry of dental implants are responsible for the transmission of load at various implant and bone interfaces, thus affecting the long-term outcomes of osseointegration [16,17,18].

In the literature, most of the studies have been conducted on CP-Ti disc samples to show the effect of different acids and their process parameters (i.e., concentration, duration, and temperature on surface topography and chemistry [8,19,20,21,22,23,24]), although the SLA surface has been used by various dental implant manufacturers where those implants have been characterized by various researchers [25,26].

However, to the best of our knowledge, no studies have documented the simultaneous effect of acid etching on the thread geometry of the Ti6Al4V ELI alloy dental implant product, its surface properties, and cytocompatibility. Moreover, fewer studies have been conducted on the SLA surface of grade 5 titanium alloys, but the effect of the etching parameter has not been discussed with regard to the surface morphology [27,28]. Since the Ti6Al4V ELI alloy is also used for the fabrication of dental implants by various manufacturers and possesses a heterogeneous composition a compared to CP-titanium, SLA surface fabrication on this alloy also needs the optimization of the acid etching process parameters. Therefore, this research endeavor aims to provide insights into the effect of the acid etching process parameters on the Ti6Al4V ELI alloy dental implants’ surface physicochemical properties and cytocompatibility, along with the optimization of micron- and nano-level topography on a patented dental implant system [29].

## 2. Materials and Methods

Dental implants (n = 57) and titanium discs (n = 63) were machined from a Ti6Al4V ELI (ASTM grade 23) 5 mm diameter rod in a 10-axis Turn-Mill machine, as shown in Figure 1. These machined implants and discs were cleaned in a mild detergent (Omega Supreme USA) and volatile solvent (3M™ Novec™ Engineered Fluids, India) in an ultrasonic cleaner (Crest ultrasonics, India) to remove machining chips as well as oil and any organic contaminations. Implants were blasted with large grit alumina particles (250–400 microns) in a sandblaster to obtain the roughness (Ra) of 1.5–2 microns. After sandblasting, the samples were cleaned in an ultrasonic cleaner (Crest ultrasonics, India) in deionized (DI) water to remove any loosely embedded blasting particles. After cleaning, samples were dried in a vacuum oven at 60–80 °C. Samples were further classified into control and experimental groups. Acid etching was carried out on these samples in HF, HCl, and H_2_SO_4_ (CDH, India), as per Table 1. Figure 2 illustrates the outline of the proposed study parameters, surface characterization, and cytocompatibility methods. The nano- and micro-SLA surfaces on the titanium dental implants were optimized by conducting the first etching in HF for 30 s without heating, and a second etching in a mixture of H_2_SO_4_ (96%) and HCl (37%) in a 1:2 ratio by volume, heated at a temperature of 80 °C for 3 min followed by 60 °C for 2 min under agitation. Post etching cleaning was done in hot deionized water in an ultrasonic bath. This optimized surface was studied in detail and compared with the acid-etched surface without heating the acid solution, as shown in Figure 2.

The surface morphology of these experimental and control implants was studied by a field emission scanning electron microscope (FESEM, FEI Quanta 200F, The Netherlands) at a 20 kV acceleration voltage, and vacuum pressure was maintained below 1 × 10^−5^ torr.

A 3D optical profilometer for surface roughness testing (KLA Tencor, Lengen, Germany) was used to analyze the surface roughness of the titanium specimens. White light optical interferometry (a non-contact method) was used to measure in a vertical scanning interferometry mode at a magnification of 20×. A 250 × 250 µm^2^ area was scanned at three random sites and Ra was calculated by inbuilt software and parameters were recorded after applying a Gaussian filter. 3D images were also recorded from the virtual reconstruction at a magnification of 20×. The arithmetical mean roughness (*R*_a_) from three areas was averaged out and standard deviations were calculated.

A vision microscope (Banbros, India) was used to study the optical profile of the dental implant thread geometry. Surface hydrophilicity was studied after being washed with distilled water in an ultrasonic cleaner for 30 min and vacuum dried. The wettability of the titanium disc specimens was examined by an optical contact angle (C.A.) measuring device (Kruss, Germany) using 2 μL of distilled water.

Surface chemical composition was evaluated using X-ray photoelectron spectroscopy (XPS, Multi-Technique Scanning XPS Microprobe, Versa Probe III USA). This method allows for the qualitative and quantitative estimation of all elements present in the sample, except hydrogen (H) and helium (He) and generates a photoelectron spectrum, with characteristic peaks for each element. Implants (as received without sputtering) were mounted on a 60 mm sample plate using double-sided nonconductive adhesive tape, with tape only contacting the support screw. Spectra were collected using a dual-beam neutralization system, consisting of low energy (~1 eV) electrons and low energy (~8 eV) argon ions. The area located in the middle part of the implant surface was randomly selected to evaluate the surface chemical composition. A monochromatic Al K_α_ (1486.6 eV) X-ray source was used with a 45° takeoff angle of photoelectrons. A 100 µm diameter 25 W X-ray beam was used at 280 eV analyzer pass energy with 1 eV/step for survey scan. For narrow scans, a 100 µm diameter 25 W X-ray beam was used at 55 eV pass energy with 0.1 eV/step for Ti 2p, C 1s, O 1s, Al 2p, V 2p, N 1s, and 69 eV pass energy with 0.125 eV/step Ca 2p, S 2p, Si 2p, F 1s, and Fe 2p.

X-ray diffractometry (XRD) measurements were carried out on the three different regions of the implants in a D/Max Ultima X-ray diffractometer (Rikagu, Tokyo, Japan). Cu-Kα1 radiation was used in a 20–80° two theta (2θ) angle range with a grazing incidence of 3 degrees.

Human osteosarcoma MG-63 cell lines were procured from the National Center for Cell Science (NCCS), Pune, India. Cells were cultured in Dulbecco’s modified Eagle medium (DMEM) supplemented with FBS (10%) and penicillin/streptomycin (1%) in a CO_2_ incubator at 37 °C. Titanium samples (2 × 5 mm diameter) were packed in class 1000 cleanroom and sterilized by gamma radiation (25 Gray) at the Shri Ram Institute of Industrial Research, Delhi, India. Cell viability and proliferation were studied by the MTT (3-(4,5-dimethyl thiazolyl-2)-2,5-diphenyltetrazolium bromide) assay. Samples were seeded with the cell density of 5000 cells/cm^2^ in 24-well plates and the MTT assay was conducted after three and seven days of culture. Commercially available MTT from Sigma was used to study the MG-63 cell proliferation on different titanium surfaces. After three and seven days of cell culture, discs were rinsed with PBS and transferred to 96-well plates and incubated with the MTT reagent for 4 h in an incubator. There was a formation of a purple color formazan crystal due to cleavage of the tetrazolium ring by the mitochondrial dehydrogenase of live cells. After incubation, dimethylsulfoxide (DMSO) was added to dissolve the formazan crystals and the optical density of the resulting purple solution was measured spectrophotometrically at a 574 nm wavelength.

FESEM evaluation of cell adhesion and morphology was conducted after 4 and 24 h of the cell culture period. For FESEM, disc samples were fixed with 4% formaldehyde after 4 and 24 h of culture, washed with phosphate buffer solution (PBS) and then sequentially dehydrated in a series of alcohol (50–100%). After gold sputter coating, FESEM (FEI Quanta 200F, The Netherlands) was used to evaluate cell adhesion and proliferation on the disc surface.

Live and dead cell staining was done with Live/Dead Cell Double Staining Kit (Sigma-Aldrich, India) after 24 h of culture duration. This kit contains Calcein-AM, which stains viable cells green and propidium iodide (P.I.) solutions stain dead cell nuclei in red. Cell culture media were removed from the incubated disc samples without agitating the well plate followed by incubation with 2 mL PBS containing 2 µM of Calcein AM and 4 µM of P for 20 min. The sample surface was evaluated under a FLUOVIEW FV1200 confocal laser scanning microscope following staining [30].

Immunofluorescence staining with rhodamine-phalloidin and 4′,6-diamidino-2-phenylindole (DAPI) was done 24 h after cell culture. Disc samples were washed with PBS and then fixed with 4% formaldehyde for 20 min. Again, the samples were washed with PBS and blocked with the 1% BSA solution for 1 h, and then disc samples were again washed with PBS and stained with rhodamine-phalloidin for 20 min and DAPI for 5 min. Cell morphology was seen under a FLUOVIEW FV1200 confocal laser scanning microscope.

All the experiments were conducted in a triplicate manner. Experimental data are presented with the mean and standard deviation (SD) and the results were evaluated statistically using two-way ANOVA test and Bonferroni post-hoc test (OriginPro 2016) between two groups (* *p* < 0.05).

## 3. Results

### 3.1. Effect of Different Etching Parameters on SEM Topography

Figure 3A,B show the SEM topography of sandblasted dental implant etched with HF (15% HF *w*/*v* for 30 s at room temperature). The sandblasting of the dental implant surface with alumina particles (250–400 microns) resulted in a surface topography with craters and pits of 10–40 microns. This surface topography can be depicted as rough but not porous. The beta phase was observed in the form of crystals on this surface, as they become prominent due to the selective dissolution of the alpha phase by HF. The size of beta crystals was found in the range of 0.5–2 microns.

Figure 4 shows the surface topography after the second etching (subsequent to the first etching in HF) in a mixture of H_2_SO_4_ + HCl solution at room temperature with different durations and temperatures. It is demonstrated in Figure 4A that when etching was done for 1 min duration, there was no significant difference in topography compared to the etched surface with HF only. When the etching duration was increased to 5 min, the formation of ridge and groove types of topography started (Figure 4B). Furthermore, when the duration was extended to 10 min of acid etching, the grooves and ridges became more prominent (Figure 4C). However, the presence of beta phase crystals was observed on all surfaces irrespective of the duration of etching times, although a relatively slight reduction in the number of crystals could be seen in these SEM images.

Figure 4D–F represents the SEM micrograph of the dual acid-etched implant surface after the second etching in a H_2_SO_4_ + HCl mixture at different temperatures (at 40 °C, 80 °C, and 120 °C) for a duration of 3 min. There is a partial etch groove and ridge type of topography with retained beta crystal when the second etching was done at 40 °C, as shown in Figure 4D. A porous topography was obtained when etching was done at 80 °C with uniform etching of both alpha and beta phases, as depicted in Figure 4E. At 120 °C, an entirely different etched topography was observed in the form of globules with grooves. These globular structures were not porous; instead, they showed a multidirectional rough and groovy topography together, suggesting a higher etch rate (Figure 4F). Figure 4G–I represents the SEM micrograph of the dual acid-etched implant surface after the second etching in the H_2_SO_4_ + HCl mixture at a higher temperature (80 °C). Here also, experiments were conducted for three-time parameters (i.e., 1 min, 5 min, and 10 min). When etching was conducted for 1 min, although there was the formation of distinct porous microstructure, the formed pores were superficial in morphology and remnants of beta crystals were observed on the surface (Figure 4G). When the duration of acid etching was increased to 5 min, a three-dimensional porous structure was obtained, as depicted in Figure 4H. This surface was highly porous and the beta crystal phase was not seen on the surface. Furthermore, when the duration of etching was increased to 10 min, a loss in the porous microstructure was noticed due to further etching action of the acids, leading to flattening of ridges resulting from sandblasting (Figure 4I).

For the optimization of both nano- and micron-topography, the second etching of the final surface was done initially for 3 min at 80 °C, followed by 2 min at 60 °C. This results in a porous surface with a pore size distribution in the micron, sub-micron, and nano range. The larger pores were in the 2–5 micron range and the smaller pores were in the 50–200 nm range, as shown in Figure 5.

Figure 6 shows the comparative topography of the present SLA surface on the alloy surface and SLA surface of the commercial marketed implant. The commercial implant surface showed a nearly uniform topography of fine cone like projections in the micron size range, as shown in Figure 6B. However, as described above, the alloy SLA surface exhibited a porous topography with a pore size ranging from the nano to micron size and had smaller pores inside the larger pore, as depicted in Figure 6A.

### 3.2. Effect of High Temperature (HT) on Thread Geometry of the Dental Implants

Optical profilometry images of dental implants (machined and sandblasted conditions) under a vision microscope are shown in Figure 7. There is a slight rounding of micro threads in an acceptable range as depicted in Figure 7C after sandblasting compared to machined implants when blasted with an appropriate combination of grit size, the pressure of blasting, and duration of blasting to achieve the desired surface roughness in the range of 1.5 to 2 microns. When acid etching was carried out with H_2_SO_4_ + HCl for 10 min at high temperature, the depth of the micro thread was reduced to less than half of the original thread depth, which is unacceptable, as shown in Figure 7G. Additionally, it was observed that there was almost no change in thread depth compared to the blasted implant when acid etching was done for 5 min at 80 °C.

### 3.3. Surface Roughness and Hydrophilicity

The 3D image and profile images of machined, sandblasted, sandblasted and acid-etched surfaces at RT and HT are shown in Figure 8A–D. Machining grooves and ridges could be observed on the machined titanium surface (Figure 8A), which corresponds to sharp peaks and valleys in the profile image. However, the overall average roughness of machined samples was below 0.5 microns. Figure 8B represents the 3D image of the sandblasted sample where the formation of the crater and ridge could be observed, and the profile image of this surface was very irregular due to non-uniform removal of material due to the sandblasting procedure. Figure 8C,D shows the 3D image of sandblasted and acid-etched surfaces at RT and HT, respectively. The superimposition of submicron roughness profile over the roughness resulted from sandblasting, and there was an overall reduction in peak height due to the etching process on the sandblasted surface. However, the roughness profile of thee RT etched surface was slightly less regular compared to the HT etched surface. The Ra values of the sandblasted and acid-etched surfaces were in the range of 1.6 to 2.1 microns, as shown in Table 2.

The surface hydrophilicity of the etch surfaces was determined by measuring the static contact angle by the sessile drop method. The machined surface showed an obtuse contact angle (110°), suggesting a hydrophobic surface. Acid-etched samples at room temperature samples had a 70° contact angle, and sample etched at a higher temperature showed a more hydrophilic surface with a contact angle of 40°.

### 3.4. XPS and XRD Analysis of Surfaces

XPS analysis was performed to study the surface elemental compositions and oxidation state of elements present on the surface. The elemental composition of implant surfaces etched at room temperature and high temperature is shown in Table 3. XPS survey spectra showed comparative peaks of different elements on the RT etched surface and HT etched surface. There was an increase in the intensity of C, O, and F peaks on the RT acid-etched surface, whereas the HT acid-etched surface showed the increased intensity of the Ti peak. Detailed elemental scans of Ti, Al, V, C, O, and F with deconvoluted spectra were evaluated and reported in Table 4 with the binding energy of all elements. It was found that the titanium was present in the Ti(III) and Ti(IV) oxidation state on the RT etched surface while on the HT etched surface, metallic titanium peaks were also detected at a 459.63 eV binding energy, as shown in Table 4. Detailed narrow scan of Al and V also showed metallic peaks on the HT acid-etched surface, suggesting a thin oxide layer on the HT etched implant.

There was a significant decrease in carbon signals at the HT acid-etched surface compared to RT. Binding energy at 284.79 corresponded to C of C–H (hydrocarbon), 286.27 corresponded to C–O, and 288.78 to O–C=O. Oxygen spectra consisted of four components TIO_2_, TiOH, O–C=O, and H_2_O_._ In the case of the O1s spectrum, the signal corresponding to oxygen atoms of TiO_2_ was increased at the RT surface, suggesting a thicker oxide layer. However, the ratio of TiOH/TiO_2_ was increased on the HT etched surface compared to the RT surface, hence a greater hydrophilic surface. High-resolution spectra of F on the RT etched surface showed the presence of AIF_3_ and TiF_6_, whereas the HT etched surface showed only a small peak of TiF_6_. The presence of AIF_3_ and TiF_6_ on the RT etched surfaces might explain the slower etch rate of the second etchant (HCl + H_2_SO_4_) at room temperature.

XRD graph of both the surfaces is shown in Figure 9. Peaks at HT acid-etched samples were nearly similar to the machined surface [31,32]. The intensity of βTi 110 peak at 38.43° theta angle was higher on the RT acid-etched surface compared to the HT acid-etched surface. However, the αTi 101 peak intensity at 40° theta angle was reduced on the RT acid-etched surface compared to the HT etched surface. Hence, the XRD cross confirmed the SEM finding that the RT acid-etched surface had more β phases exposed on the surface. The XRD results also showed that there were no TiH_2_ peaks on both the RT and HT acid-etched surfaces. Moreover, anatase phase peaks were not detected on both surfaces.

### 3.5. In Vitro Biocompatibility Study of Two Implant Surfaces

Figure 10 shows representative SEM images of cells adhered to the different surfaces after 24 h of cell culture. There was around a two times increase in the number of adhered cells on the RT and HT etched surface compared to the machined surface. On the machined samples, cells exhibited rounded cell morphology. Cells showed rounded to polygonal morphology with thin cell periphery on the RT etched surface cells. In contrast, the HT acid-etched surface demonstrated round to elongated morphology and more extended multiple filopodia, depicting more focal contacts. In high magnification, the extension of filopodia to the porous topography in the case of HT can clearly be seen.

In Figure 11, the SEM morphology of cells is shown after three days of cell culture. All three surfaces displayed a well-spread morphology. However, the machined on showed no filopodia, whereas the RT etched surface exhibited small filopodia extension and the HT etched surfaces showed the communication of cells through larger filopodial extensions, as shown in Figure 11F.

Cell proliferation was studied on days 3 and 7 of cell culture. Figure 12 shows the bar diagram of relative optical density of MG 63 cells on the machined, RT, and HT etched surfaces estimated by the MTT assay. There was a statistically significant increase in the proliferation on the HT etched surface (* *p* < 0.05) compared to the machined surface on day 7.

Figure 13 shows the fluorescent images of live and dead after 24 h of cell culture on the machined and different etched surfaces. Calcein stains live cells as green and propidium iodide (PI) stains dead cells in red. All surfaces showed good cell viability, suggesting a good cytocompatibility of the surfaces. However, cells had more spreading and filopodial extensions on a HT etched surface, similar to the results obtained from FESEM. Figure 14 shows the fluorescent microscope images of cells adhered to different surfaces where the cell cytoskeleton (actin filament) was stained by rhodamine-phalloidin and DAPI stained cell nucleus. There was an increase in cell numbers on the HT acid etched surface compared to the RT etched and machined. On the RT etched surface, actin filaments were spread in irregular directions whereas on the HT etched surface, they run along the long axis of the cell. Cells had a more spread morphology on the HT etched surface compared to the RT etched and machined surface.

## 4. Discussion

In the present study, the acid etching procedure was optimized for the Ti6Al4V ELI dental implant surface to achieve a micro- and nanotextured surface with improved hydrophilicity and surface roughness. HF was used to remove the passive oxide layer as it possesses a natural tendency to react with the titanium present in the TiO_2_ layer. However, as the etch rate of HF was very high and uncontrollable, HF alone cannot produce the desired surface microtopography. Moreover, there was a preferential etching of the alpha phase by HF in the case of TI6Al4V alloy, resulting in the residual beta phase, as shown in Figure 2. The preferential etching of the alpha phase over the beta phase was due to the fact that the beta phase had a higher concentration of vanadium, while the alpha phase consists of aluminum and the standard potential of vanadium was higher than aluminum, which is considered more novel and more resistant to etching [31,32]. The presence of the beta phase residue may result in ion leaching from the implant surface in vivo due to corrosion and in the short- or long-term may lead to osteolysis, bone loss, and implant failure [33,34,35,36,37]. Therefore, second etching is necessary to remove these unstable crystals and to produce the desired topography. H_2_SO_4_ and HCl are the most widely used etchant combination for titanium dental implants [19,22,38,39,40,41,42,43].

In this study, instead of the sequential use of these acids, the mixture of H_2_SO_4_ + HCl acids was used to reduce the number of etching steps (as there is no reaction between these two acids and these can be mixed). The etching of Ti6Al4V at room temperature by H_2_SO_4_ and HCl takes a very long time to achieve a micro-texture because of the slower etch rate of titanium at room temperature by these acids [44]. The present study reports that even after 10 min of etching, the desired micro and nanotopography could not be achieved, although roughness and wettability were in the acceptable range. Moreover, the extension of etching duration for a longer time may lead to the formation of titanium hydride (TiH_2_), leading to the brittleness of the surface layer, resulting in ion leaching creating short-term pain and inflammation and bone loss in the longer term as understood from the literature [9,45,46,47,48,49,50]. In the present study, XRD analysis showed no TiH_2_ peak on both room temperature and high temperature etch surfaces. This can be explained on the basis that titanium alloy possesses phase α and β, CP-Ti only has α phase and the β phase can accommodate more H_2_ generated during the process of etching due to the presence of a body centered cubic (bcc) structure. However, the extension of etching duration might lead to the excess generation of H_2_, hence precipitation of TiH_2_ needles [25]

The present investigation evaluated the surface chemical composition with XPS, a suitable demonstrated technique in the literature to quantify the elemental chemical composition present on the surface [48]. The XPS study result showed an increase in the intensity of C, O, and F peaks on the room temperature samples, suggesting their higher concentration since these elements can either be absorbed from the atmosphere or can be incorporated during the manufacturing process itself. The presence of the increased amount of C–H on the room temperature surface indicates an increase in the amount of hydrocarbon contamination. The machining and sandblasting procedure can contaminate the oxide layer, which needs to be removed during the subsequent acid etching process [48].

The presence of oxygen was evident on both surfaces since, in addition to being present in the atmosphere, it was part of the metallic oxides used during the sandblasting processes. However, etching with HF removed the oxide layer, but again, as the process was not done under an inert atmosphere and a second etching was carried out at room temperature, there was again an increase in oxide layer thickness on the room temperature acid-etched surfaces while the high temperature acid-etched surface displayed metallic pecks of T, V, and Al, suggesting a thin oxide layer and the capability of high temperature acid to remove the oxide layer during process. Therefore, it has been suggested that the Ti implant surface should contain a high Ti/C ratio [51,52]. The presence of a high Ti/C ratio and Ti–OH groups is an important feature because it is strongly related to surface wettability, as observed in the present research work and also in the literature [50]. During the manufacturing process of the implants due to contact with the organic lubricating oil, a large number of hydrocarbons are easily attached on the surface, thereby increasing the carbon content and reducing the property of Ti [48].

Biocompatibility of the optimized AE surface was greater, which can be attributed to the greater hydrophilic surface, less carbon contamination, higher Ti/C ratio, increase in TiOH group on the Ti surface, and the presence of nanotopography along with microtopography. Surface micro- and nanotopography, surface roughness, and hydrophilicity enhanced the process of osseointegration [51,52] as osteoblasts (key player cell for bone deposition) were stimulated by the microenvironment regulating bone remodeling on rough micro and nanotextured surfaces. These surfaces modulate the process of protein absorption, cell recruitment, cell adhesion, the formation of focal contact, cytoskeletal organization, proliferation, gene expression, and cell differentiation [53,54,55,56].

Saulacic et al. [57] stated that acid etching of the Ti6Al4V alloy is typically not an appropriate treatment due to its biphasic nature. Acid etching leads to an enrichment of the surface with the vanadium rich β Ti phase. However, the present research helped in partially disapproving this statement by providing proof of concepts. According to the present research work, it is the parameters of acid etching and type of acid that can be used to provide the acid-etched surface that is similar or even more porous than the CP-Ti surface without enriching the surface with the β Ti phase. As parametric studies of acid etching on alloy surfaces are lacking in the literature and most of the implant manufacturing companies do not disclose the exact etching process, there was a general agreement not to perform acid etching process on alloy surface implants.

Recently, Budei et al. [23] compared the morphology of different dental implant systems and concluded that different companies used either Ti Grade 4 or the Ti6Al4V-ELI alloy. Although they studied the morphology, they did not discuss this in terms of the remaining β Ti phase on the surface of the alloy implants. However, they concluded that the surface morphology was different for each implant system. Similar to our observations, Tait et al. [45] observed the SEM morphology of the osteotite surface and concluded that the acid etching was not sufficient to dissolve both α and β phases of the alloy; hence, the deposition of sharp white grains of the β phase look like alumina particles rich in vanadium.

In the present study, the second step of acid etching was done in a mixture of H_2_SO_4_ + HCl at 60–80 °C temperature, and it was found that 5 min of duration was sufficient to achieve the controlled nano- and microtopography devoid of beta crystals. These surfaces were more hydrophilic, with fewer chances of hydrogen embrittlement. The novelty of this research work includes a detailed step-by-step effect of the acid etching process was studied on the Ti6Al4V alloy dental implant surface for the first time, which was necessary to explain the previous work. This study provides a future direction for a better surface modification of the Ti6Al4V alloy implant surface by various companies. This work might further increase the popularity of using Ti6Al4V alloy implants, requiring more strength without fear of vanadium toxicity.

In short, it can be concluded that although acid etching is a necessary process on the titanium dental implant surface in order to improve its in vivo bioactivity, over-etching or under-etching of the titanium alloys may lead to a compromise in either mechanical properties, corrosion resistance, or biocompatibility. Hence, the type and concentration of acids and their sequence of process and duration must be optimized for each type of desired microstructure and material composition as well as thread type

The major limitations of the present study include the use of the MG-63 cell line for conducting a cytocompatibility study. In future work, human mesenchymal stem cells and human fetal osteoblast cells can be used. More detailed differentiation and mineralization studies and osteogenic markers studies can be carried out. Moreover, future in vivo and human clinical trials are necessary to prove the applications of the results from the present study.

## 5. Conclusions

In this study, the parameters of acid etching were optimized for dental implants made of the Ti6Al4V ELI grade. This study also concludes that the etching behavior of the titanium alloy is different than CP-Ti. Acid etching of the titanium alloy by HF is only insufficient as it preferentially dissolves the alpha phase and exposes the beta phase boundaries, increasing the susceptibility of decreasing the corrosion resistance of the alloy. Shorter duration acid etching with H_2_SO_4_ + HCl at room temperature has a minimal effect on surface topography. Even after 10 min of acid etching at room temperature, there was no significant change in topography and we were unable to remove beta crystals that might be unstable and affect the osseointegration process. Acid etching with a combination of acids at higher temperatures can produce micro- and nano-level topography in a shorter duration with a uniform etching of both alpha and beta phases. Higher temperature acid etching improves the surface roughness by superimposing the micro- and nano-roughness on the macro roughness introduced by the sandblasting procedure. High-temperature acid etched surfaces are more hydrophilic and displayed an enhanced cytocompatibility to the room temperature etched surface.

## Figures and Tables

**Figure 1 materials-14-06344-f001:**
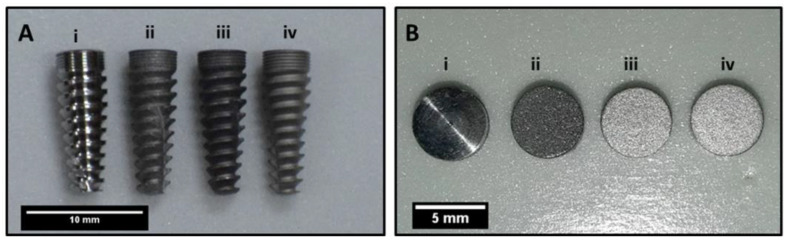
(**A**) Photograph of (i) machined, (ii) sandblasted, (iii) sandblasted and acid-etched at room temperature (R.T.), (iv) sandblasted and acid-etched at high temperature (HT) implants. (**B**) Disc samples in similar processing conditions.

**Figure 2 materials-14-06344-f002:**
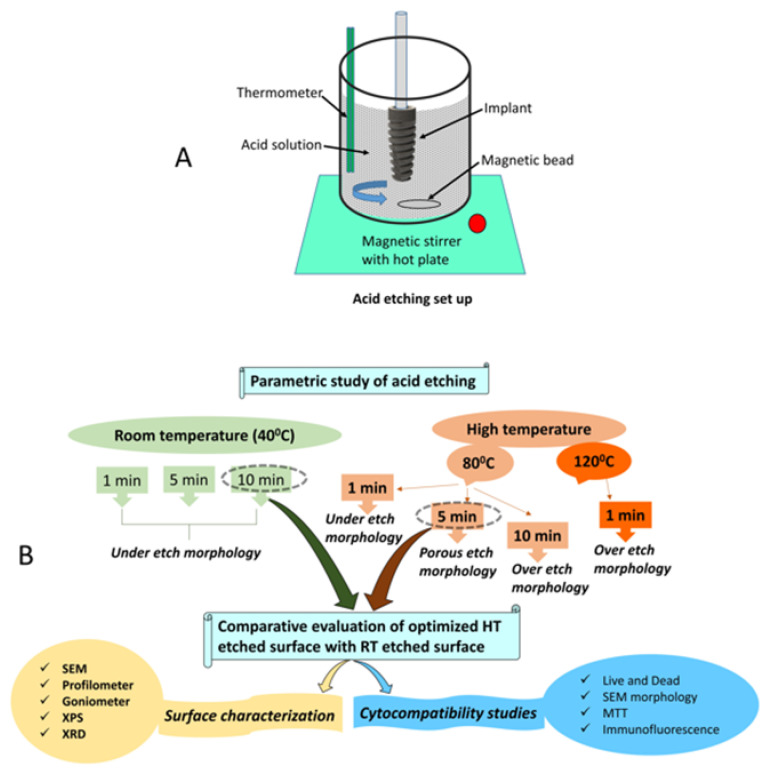
(**A**) Acid etching set- up. (**B**) Schematics representing the outline of the study and showing acid etching parameters, characterization methods, and cytocompatibility studies.

**Figure 3 materials-14-06344-f003:**
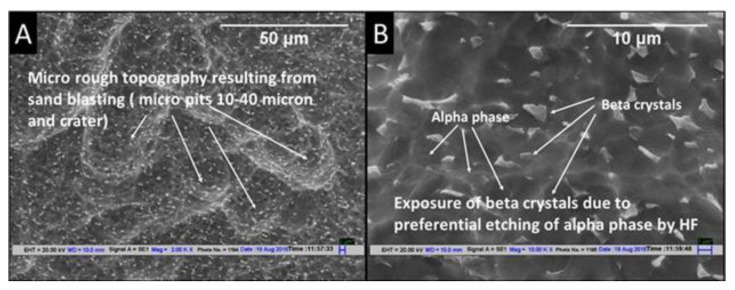
SEM images of sandblasted dental implant acid-etched with HF for 30 s at room temperature at (**A**) 2000× and (**B**) 5000×.

**Figure 4 materials-14-06344-f004:**
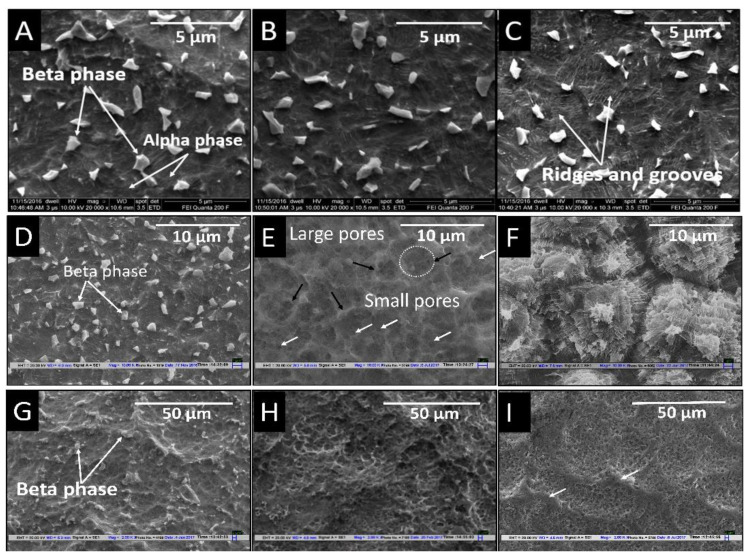
SEM images of sandblasted dental implant acid-etched with HF for 30 s (RT) and second etching with HCl + H_2_SO_4_ at room temperature (**A**–**C**) for a duration of (**A**) 1 min, (**B**) 5 min, (**C**) 10 min; (**D**–**F**) second etching with HCl + H_2_SO_4_ for a duration 3 min at (**D**) 40 °C, (**E**) 80 °C, and (**F**) 120 °C; and (**G**–**I**) second etching with HCl + H_2_SO_4_ at HT for the duration (**G**) 1 min (**H**) 5 min; and (**I**) 10 min.

**Figure 5 materials-14-06344-f005:**
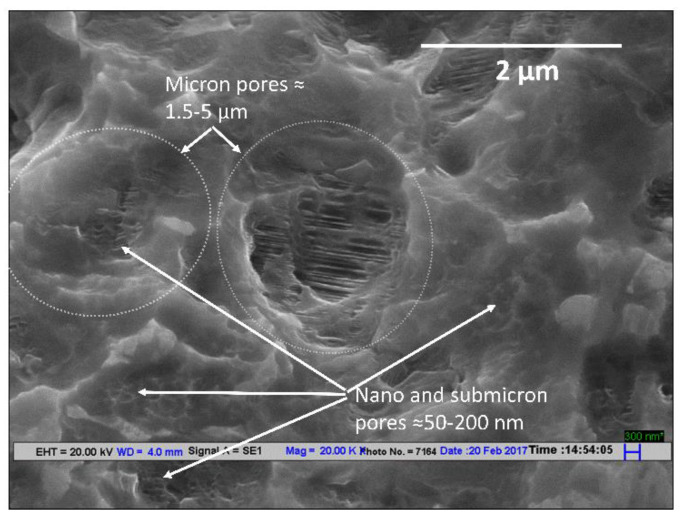
SEM images of sandblasted dental implant acid-etched with HF for 30 s (RT) and second etching with HCl + H_2_SO_4_ at HT for a duration of 5 min at 20,000× magnification.

**Figure 6 materials-14-06344-f006:**
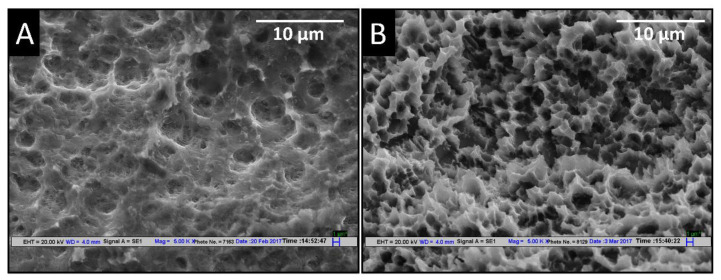
Comparative SEM images of (**A**) SLA surface of Ti6Al4V ELI grade alloy etched at 80 °C for 5 min and (**B**) SLA surface of Neobiotech commercial dental implant.

**Figure 7 materials-14-06344-f007:**
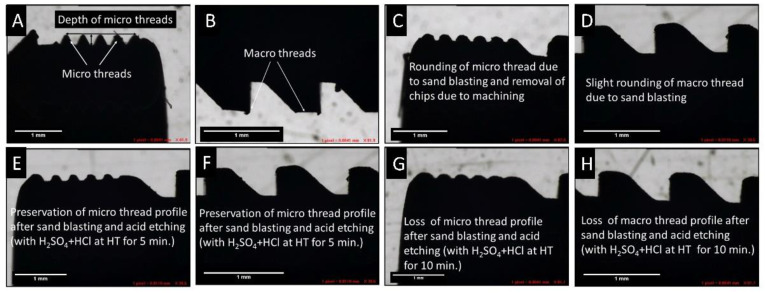
Profile images of dental implants under vision microscope: (**A**,**B**) machined implant, (**C**,**D**) sandblasted implant, (**E**,**F**) etched at HT for 5 min, (**G**,**H**) Etched at HT for 10 min.

**Figure 8 materials-14-06344-f008:**
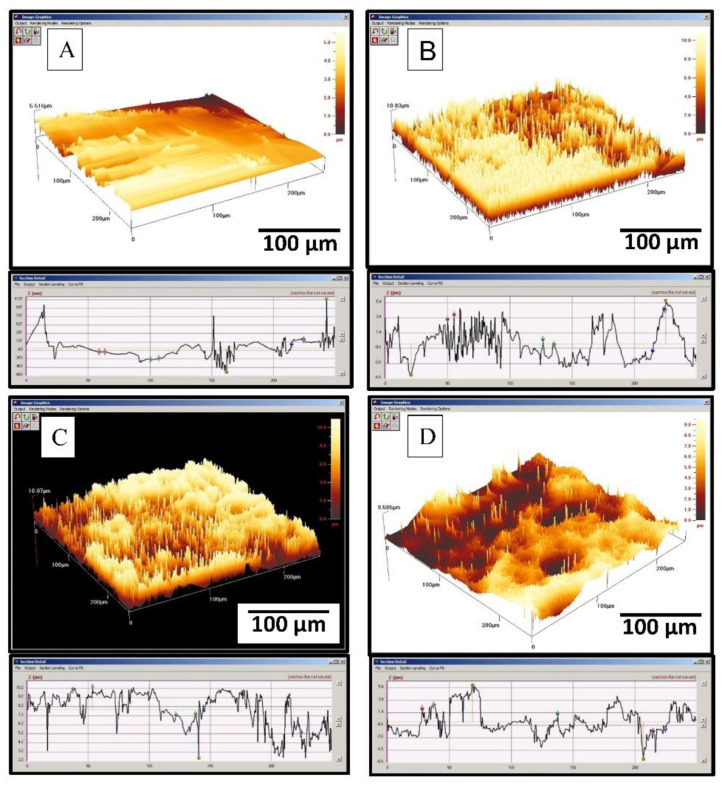
Ra value measurement: (**A**) machined, (**B**) sand blasted, (**C**) sand blasted and acid-etchedat RT (10 min), and (**D**) sand blasted and acid-etched at HT (5 min.).

**Figure 9 materials-14-06344-f009:**
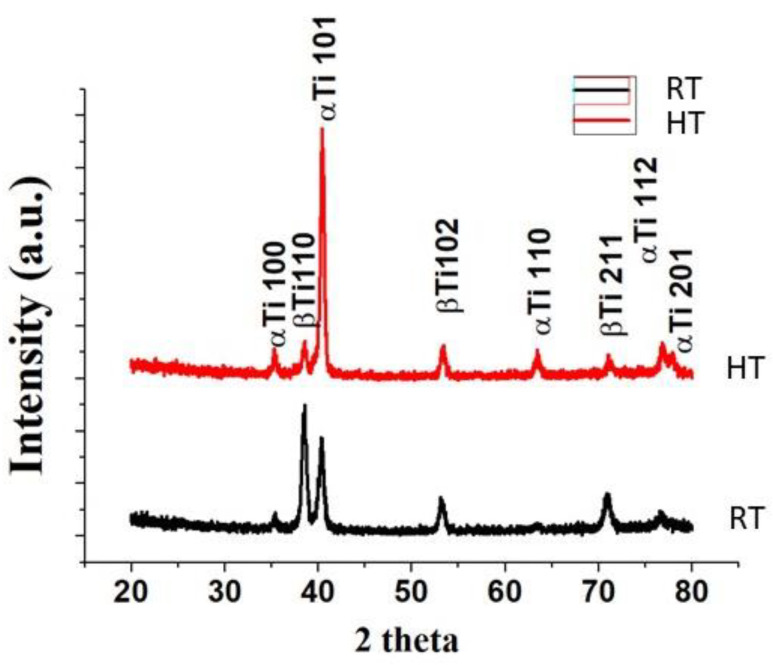
XRD diffractogram of the HT and RT etched surface of the Ti6Al4V alloy.

**Figure 10 materials-14-06344-f010:**
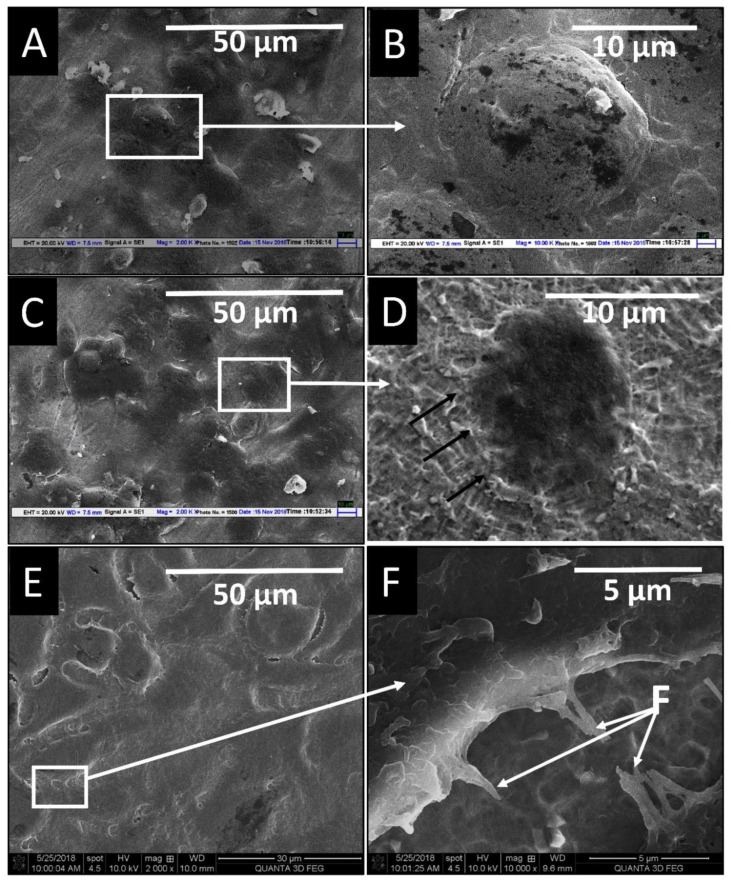
SEM images of MG 63 cells on machined (**A**,**D**), RT etched (**B**,**E**), and HT etched (**C**,**F**) surfaces after 24 h of culture (F—filopodia).

**Figure 11 materials-14-06344-f011:**
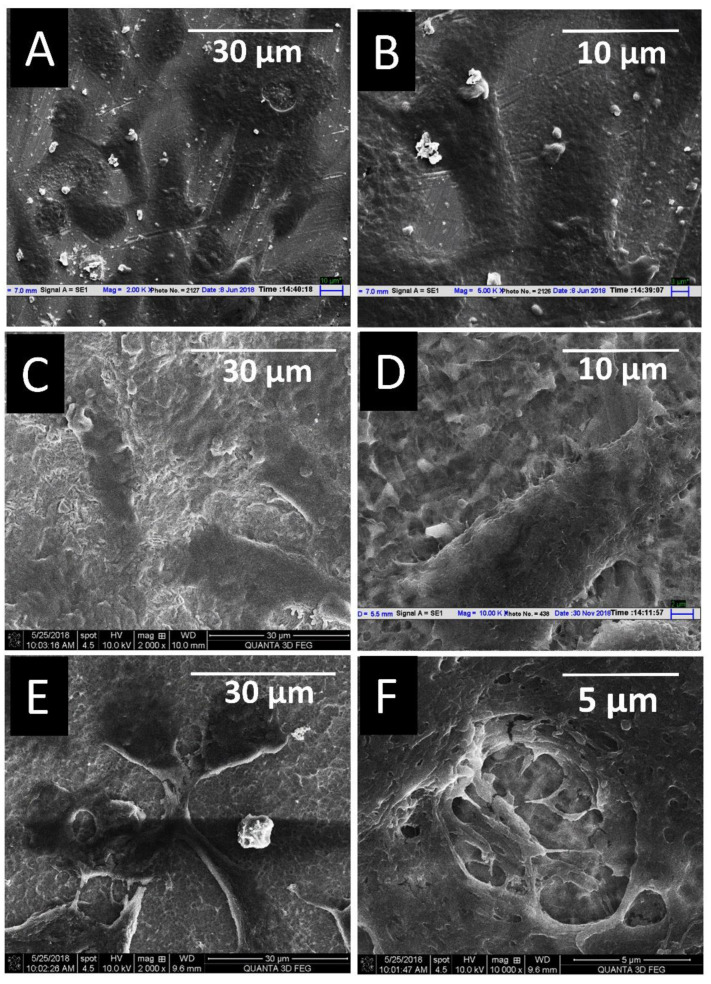
SEM Images of MG 63 cells on machined (**A**,**D**), RT etched (**B**,**E**), and HT etched (**C**,**F**) surfaces after three days of culture.

**Figure 12 materials-14-06344-f012:**
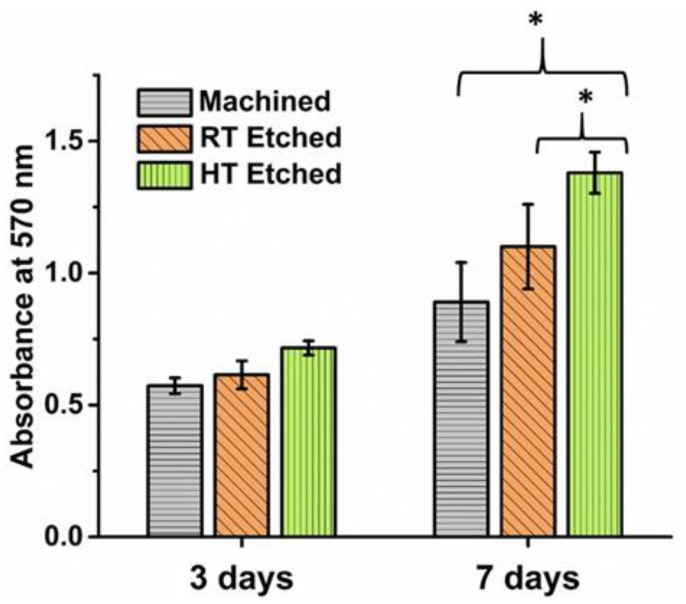
Bar diagram showing the cell proliferation of MG 63 cells on the control and experimental surfaces at three and seven days of cell culture by the MTT assay (* *p* < 0.05).

**Figure 13 materials-14-06344-f013:**
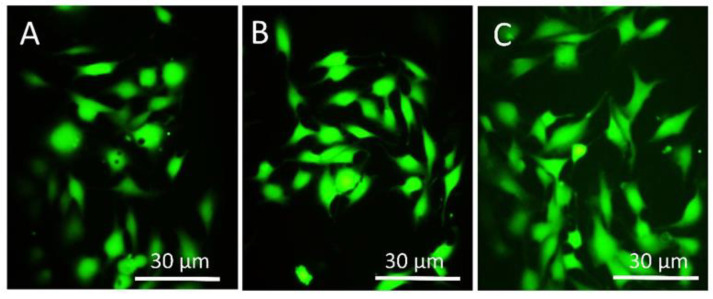
Confocal images of MG63 cells on the machined (**A**), RT etched (**B**), and HT etched (**C**) surfaces stained with Calcein green and PI after 24 h of culture.

**Figure 14 materials-14-06344-f014:**
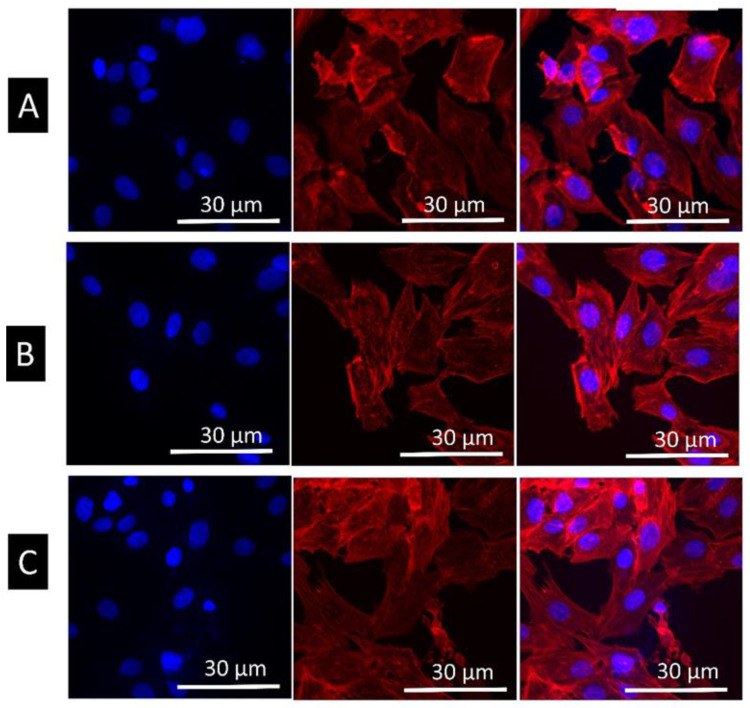
Confocal images of MG63 cells on machined (**A**), RT etched (**B**), and HT etched (**C**) surfaces as cytoskeleton stained with rhodamine-phalloidin and nucleus stained with DAPI after 24 h of culture.

**Table 1 materials-14-06344-t001:** Experimental parameters used in the study.

Acid Etching Group		Etching Solution	Temperature	Time
Single Step		HF Only	R.T.	30 s
Double step (Room temperature-RT)	*A*	HF (1st etching) H_2_SO_4_ +HCl (2nd etching)	RT RT	30 s 1 min
	*B*	HF (1st etching) H_2_SO_4_ + HCl (2nd etching)	RT RT	30 s 5 min
	*C*	HF (1st etching) H_2_SO_4_ + HCl (2nd etching)	RT RT	30 s 10 min
Double step (Different temperatures)	*A*	HF (1st etching) H_2_SO_4_ + HCl (2nd etching)	RT 40 °C	30 s 5 min
	*B*	HF (1st etching) H_2_SO_4_ + HCl (2nd etching)	RT 80 °C	30 s 5 min
	*C*	HF (1st etching) H_2_SO_4_ + HCl (2nd etching)	RT 120 °C	30 s 5 min
Double step (High temperature-HT)	*A*	HF (1st etching) H_2_SO_4_ + HCl (2nd etching)	RT 60–80 °C	30 s 1 min
	B	HF (1st etching) H_2_SO_4_ + HCl (2nd etching)	RT 60–80 °C	30 s 5 min
	C	HF (1st etching) H_2_SO_4_ + HCl (2nd etching)	RT 60–80 °C	30 s 10 min

**Table 2 materials-14-06344-t002:** Ra Value of the different experimental surfaces of the dental implant.

Surface Type	Ra in µm	Contact Angle in Degree
Machined	0.357 ± 0.132	110 ± 5
Sandblasted	2.1 ± 1.1	100 ± 2
Sandblasted and acid-etched at RT (10 min)	1.7 ± 1.2	70 ± 7
Sandblasted and acid-etched at HT (5 min)	1.6 ± 1.2	40 ± 3

**Table 3 materials-14-06344-t003:** Elemental composition (in %/) of the implant surface after RT and HT acid etching treatment as evaluated by XPS survey.

Elements	C	O	TI	AL	V	F
RT Ac-id-Etched Implant	46.3	39.8	9.1	1.3	0.1	3.4
HT Ac-id-Etched Implant	30.9	28.6	39	0.8	0.1	0.6

**Table 4 materials-14-06344-t004:** XPS binding energies of Ti2p, Al2p, V2p, C1s, O1s, and F1s at the surface of the titanium surface etched at room temperature and high temperature.

	Room Temperature (RT) Acid-Etched Titanium Alloy Dental Implant Surface			High Temperature (HT) Acid-Etched Titanium Alloy Dental Implant Surface		
	Peak Position	% Area	Oxidation State	Peaks Position	% Area	Oxidation State
**TI**	453.48	0.00	Ti(0) 2p_3/2_ (Ti metal)	453.43	4.18	Ti(0) 2p_3/2_ (Ti metal)
	455.22	0.00	Ti(II) 2p_3/2_ Ti-O	454.99	1.28	Ti(II) 2p_3/2_ Ti-O
	457.30	2.05	Ti(III) 2p_3/2_ Ti_2_O_3_	456.98	5.61	Ti(III) 2p_3/2_ Ti_2_O_3_
	458.84	64.62	Ti(IV) 2p_3/2_ TiO_2_	458.69	55.60	Ti(IV) 2p_3/2_ TiO_2_
	459.68	0.00	Ti(0) 2p_1/2_ (Ti metal)	459.63	2.09	Ti(0) 2p_1/2_ (Ti metal)
	460.82	0.00	Ti(II) 2p_1/2_ Ti-O	460.59	0.64	Ti(II) 2p_1/2_ Ti-O
	462.90	1.03	Ti(III) 2p_1/2_ Ti_2_O_3_	462.58	2.81	Ti(III) 2p_1/2_ Ti_2_O_3_
	464.51	32.31	Ti(IV) 2p_1/2_ TiO_2_	464.36	27.80	Ti(IV) 2p_1/2_ TiO_2_
**AL**	74.76	81.61	Al_2_O_3_	71.32	11.89	Metal
	76.34	18.39	AlF_3_	74.22	88.11	Al_2_O_3_
**V**	515.85	100.00	Oxide	511.74	5.95	Metal
				515.41	94.05	Oxide
**C**	284.79	84.11	C–C, C–H	284.79	88.22	C–C, C-H
	286.27	9.61	C–O	285.92	8.77	C–O
	288.78	6.28	O–C=O	288.75	3.01	O–C=O
**O**	530.19	45.27	Metal oxide	530.17	55.68	Metal oxide
	531.83	54.73	Organic, sulfates	531.69	44.32	Organic, sulfates
**F**	684.89	73.77	Fluoride	685.01	100.00	Fluoride
	686.38	26.23	AlF_3_			

## Data Availability

Data is contained within the article or Appendix A. The data presented in this study are available in Appendix A.

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
