# Peer review of "Critical Role of Etching Parameters in the Evolution of Nano Micro SLA Surface on the Ti6Al4V Alloy Dental Implants"

_materials, 2021, doi:10.3390/ma14216344_

Round 1

Reviewer 1 Report

This submission is devoted to the study of the effect of SLA condition (time and temperature) on surface parameters and in vitro characteristics of biomedical Ti6Al4V alloy. The research topic is very relevant and promising. Despite the fact that the SLA technology has been known for a long time and is widely used to modify the surface of titanium implants, the features of etching of the Ti6Al4V alloy have not yet been studied in detail. The authors have done a lot of experimental work. Many samples have been prepared and they have been studied in detail (composition, topography, morphology, in vitro). Quite interesting results were obtained. It is also worth noting a rather detailed introduction, which describes the main problems of modifying the surface of titanium implants.

Despite a large number of advantages, the work has a lot of flaws and disadvantages:

  • The materials and methods section should be supplemented by data of…
  • What kind of “detergent and volatile solvents” were used to remove machined chips?
  • The ratio of H2SO4/H2O was 1:2. Is it volume or mass ratio?
  • There is no description of XPS and XRD experiments
  • What is the Ra (profilometry data)? Is it measured or calculated? It is necessary to describe in materials and methods.
  • The authors showed results of XPS and XRD (Table 4, fig 15,16). But they did not describe it in the text and discussed it at all!
  • The paragraph in lines 218-222 strikes me as a bit odd and probably redundant. Please check
  • By now the effect of varying SLA conditions of Ti6Al4V on composition, topography, morphology, and biomedical properties was not studied in detail. But brief search in the Scopus database showed that SLA was used for Ti6Al4V implants many times by different authors. It is necessary to compare the results of this research with the literature data.
  • It is necessary to briefly describe how the presence of the beta phase was determined from SEM images. Is there any visual difference between alpha and betta grains?
  • Some of the results presented in the main text of the manuscript are redundant. For example, Figure 13 duplicates Table 2. The XPS survey spectra do not show important information if the results from Table 3 are given.
  • Many results were obtained but they don’t discuss enough. For example phase and surface composition. The presence of metallic Ti, Al and V on the HT sample and their absence on the RT sample. In the discussion section, the authors give a lot of reasoning about important, but specific problems (fragility due to hydrogen), but little discussion of the specific results of their work and comparison with the literature data
  • What are the black and white arrows in fig5b? What do they mean
  • Figure 11 is of poor quality. It is very difficult to see scales and numbers
  • Please check the error bars in fig 13A.
  • Line 337 – “different nanotube surfaces”. What does it mean?
  • There are several typos. See lines 157, 309.

 In conclusion, I would like to note that the research has a high impact, novelty and is very interesting and important for researchers who worked on surface modification and improvement bioactivity of titanium-based medical implants. The research is of a high scientific standard, but the manuscript should be remarkably improved, completed, and revised. Therefore, I would like to recommend this manuscript for publication in the Materials after major revision.

Reviewer 2 Report

The paper is interesting and can be considered for publication in materials

however before acceptance some minor concerns are needed

1) Authors should consider the use of laser in treating implant surfaces and its potential implications. Please cite PubMed ID32859022

2) Authors should hypothesize the potential use of stem cells and their behaviour with such new implant material. Please cite PubMed ID32811413

3) Authors should discuss the possible formation of biofilm on such material. Please consider PubMed ID28750776

Reviewer 3 Report

Thank you for submitting the paper  Critical Role of Etching Parameters in the Evolution of Nano Micro SLA Surface on the Ti6Al4V Alloy Dental Implants.

Figure 1 and table 1 should be AFTER mentioned in the text, so you should change it after material and method, not before.

Mat and method: “Dental implants and titanium discs were machined from” n=? the authors shoulb clarifying this in each experiment.

Why did you selected Human osteosarcoma MG-63 cell lines?

Lot of acronym without a description before (MTT, NCCS, FESEM…)

Poorly described material and methods. How was done MTT, FESEM, live and death cell staining? Why did you selected 3 and 7 days to MTT?  And 24h to FESEM? For live and death cell, it is not better to use the anexin test?

You used machined ones as control, but control should be cells alone, without anything more.

Results: Lack of significance (p <???) add in the explanations (text), in the results section.

Discussion is too short. What is the novelty of this paper? Which are the limitations?

From sentence 386 to 397 there is not any reference in a discusion section.

References: Establish all references according to the journal. Only 6 references are from the last 5 years, this is not a updated manuscript.

In addition, in my opinion it is a difficult manuscript to follow due to the large number of figures that it has that do not continue correctly with the text when you read it.

Round 2

Reviewer 1 Report

The authors answered all the questions in detail and made the necessary corrections. I believe the manuscript can be published. However, the manuscript still contains many grammatical errors and typos. I hope that the authors and the editorial board will correct these errors. 

Author Response

Thanks for reviewing the manuscript and making us modify it in a greater way. The grammatical and typo mistakes have been corrected in the revised manuscript.

Reviewer 3 Report

The authors have followed the changes suggested and have greatly improved the article. Therefore, in my opinion, this scientific article meets the necessary criteria to be published in present form.

Author Response

The authors want to appreciate the efforts of the reviewer for providing comments during the first revision in a sequential manner. It helped a lot to make the manuscript in a publishable format.